**Data Availability Statement:** The conditions under which the data were provided do not allow for the data to be made publicly available. The data we used for this paper were third-party data and we

# Association of pre-existing comorbidities with mortality and disease severity among 167,500 individuals with COVID-19 in Canada: A population-based cohort study

Erjia Ge[1], Yanhong Li[1], Shishi Wu[1], Elisa Candido[2], Xiaolin Wei[1]*

**1** Dalla Lana School of Public Health, University of Toronto, Toronto, Ontario, Canada, **2** ICES, Toronto, Ontario, Canada

* xiaolin.wei@utoronto.ca

## Abstract

### Background

The novel coronavirus disease 2019 (COVID-19) has infected 1.9% of the world population by May 2, 2021. Since most previous studies that examined risk factors for mortality and severity were based on hospitalized individuals, population-based cohort studies are called for to provide evidence that can be extrapolated to the general population. Therefore, we aimed to examine the associations of comorbidities with mortality and disease severity in individuals with COVID-19 diagnosed in 2020 in Ontario, Canada.

### Methods and findings

We conducted a retrospective cohort study of all individuals with COVID-19 in Ontario, Canada diagnosed between January 15 and December 31, 2020. Cases were linked to health administrative databases maintained in the ICES which covers all residents in Ontario. The primary outcome is all-cause 30-day mortality after the first COVID-19 diagnosis, and the secondary outcome is a composite severity index containing death and hospitalization. To examine the risk factors for the outcomes, we employed Cox proportional hazards regression models and logistic regression models to adjust for demographic, socio-economic variables and comorbidities. Results were also stratified by age groups. A total of 167,500 individuals were diagnosed of COVID-19 in 2020 and included in the study. About half (43.8%, n = 73,378) had at least one comorbidity. The median follow-up period were 30 days. The most common comorbidities were hypertension (24%, n = 40,154), asthma (16%, n = 26,814), and diabetes (14.7%, n = 24,662). Individuals with comorbidity had higher risk of mortality compared to those without (HR = 2.80, 95%CI 2.35–3.34; p<0.001), and the risk substantially was elevated from 2.14 (95%CI 1.76–2.60) to 4.81 (95%CI 3.95–5.85) times as the number of comorbidities increased from one to five or more. Significant predictors for mortality included comorbidities such as solid organ transplant (HR = 3.06, 95%CI 2.03–4.63; p<0.001), dementia (HR = 1.46, 95%CI 1.35–1.58; p<0.001), chronic kidney disease (HR = 1.45, 95%CI 1.34–1.57; p<0.001), severe mental illness (HR = 1.42, 95%CI%, 1.12–

didn't have any special access privileges to it. We acquired the data from the ICES (www.ices.on.ca), an independent, non-profit research institute whose legal status under Ontario's health information privacy law allows it to collect and analyze health care and demographic data. Release and/or sharing of these data are not covered under our current data use agreement with ICES. Any request to the data should be made directly to the ICES in the same way as described above.

**Funding:** XW received the funding from Canadian Institute of Health Research (CIHR) https://cihr-irsc.gc.ca/ and International Research and Development Centre (IDRC) https://www.idrc.ca (439835). The funder had no role in study design, data collection and analysis, decision to publish, or preparation of the manuscript.

**Competing interests:** The authors have declared that no competing interests exist.

**Abbreviations:** COVID-19, coronavirus disease 2019; COPD, chronic obstructive pulmonary disease; CVD, cardiovascular disease; CI, confidence interval; HIV, human immunodeficiency virus; HR, hazard ratio; IQR, inter-quartile range; LTC, long-term care; OHIP, Ontario Health Insurance Plan; OLIS, Laboratories Information System; OR, odds ratio; PCR, polymerase chain reaction; SARS-CoV-2, severe acute respiratory syndrome coronavirus 2; SD, standard deviation; VOC, variants of concern.

1.80; p<0.001), cardiovascular disease (CVD) (HR = 1.22, 95%CI, 1.15–1.30), diabetes (HR = 1.19, 95%, 1.12–1.26; p<0.001), chronic obstructive pulmonary disease (COPD) (HR = 1.19, 95%CI 1.12–1.26; p<0.001), cancer (HR = 1.17, 95%CI, 1.09–1.27; p<0.001), hypertension (HR = 1.16, 95%CI, 1.07–1.26; p<0.001). Compared to their effect in older age groups, comorbidities were associated with higher risk of mortality and severity in individuals under 50 years old. Individuals with five or more comorbidities in the below 50 years age group had 395.44 (95%CI, 57.93–2699.44, p<0.001) times higher risk of mortality compared to those without. Limitations include that data were collected during 2020 when the new variants of concern were not predominant, and that the ICES databases do not contain detailed individual-level socioeconomic and racial variables.

## Conclusion

We found that solid organ transplant, dementia, chronic kidney disease, severe mental illness, CVD, hypertension, COPD, cancer, diabetes, rheumatoid arthritis, HIV, and asthma were associated with mortality or severity. Our study highlights that the number of comorbidities was a strong risk factor for deaths and severe outcomes among younger individuals with COVID-19. Our findings suggest that in addition of prioritizing by age, vaccination priority groups should also include younger population with multiple comorbidities.

## Introduction

Since the first case of the coronavirus disease 2019 (COVID-19) was reported in December 2019, the exponential growth of the pandemic has profoundly changed every aspect of human lives. By May 2, 2021, there were over 151 million reported COVID-19 cases, accounting for 1.9% of the world population, among them over 3 million died [1]. Since October 2020, COVID-19 become the third leading cause of death in the United States for adults aged 45 years or older [2], and is likely to continue to rise over the list in many countries [3].

Existing evidence from a growing body of research on the risk factors for adverse outcomes of COVID-19 revealed that old age is a strong predictor of COVID-19 mortality; in addition, male sex, hypertension, diabetes, cardiovascular disease (CVD), kidney disease, cancer and dementia are risk factors for COVID-19 mortality and hospitalization [4–10]. However, the majority of the previous studies was based on cohorts of hospitalized COVID-19 patients because electronic medical records were more accessible [5, 9, 11]. One limitation of using hospital-based cohorts is that risk factors identified from severe cases may not apply to the general population, since the majority of COVID-19 cases are in the community. Therefore, evidence from population-based cohorts is called for to provide a more comprehensive and robust analysis of risk factors that can be extrapolated to the general population. As of May 2 2021, only four studies employed cohorts containing both outpatients and inpatients in examining risk factors for COVID-19 mortality. The two studies that were conducted in the United States and the South Africa are based on public health insurance records [12, 13], which may miss a large proportion of residents with private insurance. In addition, the study in the United States did not report relative risks [13] and the South African study contained limited information on comorbidities [12]. Another study employed research network databases in the United States which had limited population coverage [10]. The study in South Korea were based on national health insurance data with a full population coverage [14], but the accuracy of

diagnoses of asthma and diabetes was as low as 50% due to false claims made for profit reasons [15]. Furthermore, all the previous studies were based on data from the early wave of the pandemic, while more cases among younger population were observed in the following waves [16, 17]. Therefore, there is an urgent need to capture the changing risk factors for COVID-19 outcomes using large and accurate population-based data to provide evidence to inform vaccination roll-out, public health and clinical responses.

The first COVID-19 case in Canada was reported in Toronto, Ontario on January 25, 2020. By May 2, 2021, there were over 1.23 million cases and 24,300 deaths reported in Canada [18]. As the largest province in Canada, Ontario reported the highest number of cases (470,465) and the second highest number of deaths (8,102) by May 2, 2021 [19]. In Ontario, health care services, including physician services, hospital care, and diagnostic testing, are universally-funded for all residents with the provincial government as the single payer. Using population-based health administrative databases, we aim to examine the associations of comorbidities with mortality and disease severity in individuals with COVID-19 diagnosed in 2020 in Ontario, Canada.

## Methods

### Study population and data sources

We conducted a population-based retrospective cohort study of individuals who had a positive test result of the novel coronavirus (SARS-CoV-2) reported through the Ontario Laboratories Information System (OLIS) from January 15 to December 31, 2020. Individuals who were not eligible for the Ontario Health Insurance Plan (OHIP) and those who were not residents of Ontario at the beginning of the study period were not included in the database. The OLIS collects testing SARS-COV-2 results based on the polymerase chain reaction (PCR) test processed at all provincial public health laboratories, hospital laboratories, and commercial laboratories in Ontario. We treated the first date of sample collection with a following positive SARS-COV-2 result diagnosed in 2020 as the diagnosing date. The study cohort derived from OLIS was linked to population-based provincial health administrative data to ascertain baseline information on socio-demographic characteristics and chronic conditions, as well as outcomes of interest. These datasets were linked using unique encoded identifiers and analyzed at ICES. ICES is an independent, non-profit research institute whose legal status under Ontario's health information privacy law allows it to collect and analyze health care and demographic data, without consent, for health system evaluation and improvement.

### Ethics consideration

The study has received ethical approval from the Office of Research Ethics at the University of Toronto (#39138). The consent form was not obtained as the data is from ICES, which is an independent, non-profit research institute whose legal status under Ontario's health information privacy law allows it to collect and analyze health care and demographic data, without consent, for health system evaluation and improvement.

### Definitions of outcomes and risk factors

The primary outcome of our study is mortality, which is defined as all-cause death within 30 days after the first positive SARS-COV-2 test. Data on individuals' mortality was retrieved from the Canadian Institute for Health Information's Discharge Abstract Database and the Registered Persons Database until January 30, 2021.

We adopted a composite severity outcome, which consists of all-cause deaths within 30 days after the first SARS-COV-2 positive test and hospitalization happened between the period of 14

days prior to and 30 days after the first SARS-COV-2 positive test, as our secondary outcome. Data on hospital admission and discharge was retrieved from the National Ambulatory Care Reporting System, and the Ontario Health Insurance Plan databases until January 30, 2021.

We extracted individuals' demographic and residential information, including age, sex, residence, income quintile, and whether the individual was resident of a long-term care (LTC) facility 90 days prior to the testing date from the database. Age was grouped as below 50, 50–59, 60–69, 70–79, and above 80 years old. Residence was distinguished between rural and urban. The income variable was neighborhood-based and determined using methods developed by Statistics Canada, where income was adjusted for household size and cost of living across the province so that each dissemination area would have 20% of its population in each income quintile. Quintile five indicated the highest income group while quintile one indicated the lowest. Each individual was assigned with the neighborhood income quintile of the dissemination area which was matched with his/her postal code [20]. LTC residence was included as risk factor because high mortality in LTC facilities was reported in several countries, including Canada [21–23]. Individuals' pre-existing comorbidities (S1 Appendix) were identified using pre-existing ICES chronic disease cohorts and data from the Discharged Abstract Database, National Ambulatory Care Reporting System, Ontario Health Insurance Plan, and the Ontario Drug Benefit Claims Database. ICES chronic disease cohorts have been derived for several chronic conditions using administrative data algorithms. The majority of these have been validated through chart review and demonstrate high sensitivity and specificity [24, 25]. The comorbidities include asthma, chronic pulmonary obstructive disease (COPD), dementias (including Alzheimer's and delirium), human immunodeficiency virus (HIV), hypertension, diabetes, chronic kidney disease, cancer, CVD (including cardiac ischemic disease, congestive heart failure, acute ischemic stroke, and hemorrhagic stroke), rheumatoid arthritis, inflammatory bowel disease, liver disease, severe mental illness (including other hospitalized mental illness excluding dementia), and solid organ transplant. A list of the case definition of the comorbidities can be found in S1 Appendix.

## Statistical analysis

We first conducted descriptive statistical analyses of risk factors, by calculating proportions, means with standard deviation (SD), and medians with inter-quartile ranges (IQRs) of the variables. To compare the differences in risk factors between the deceased and alive groups, we employed statistical tests such as student-t test, Wilcoxon rank-sum test, Chi-Square test, Fisher's exact test, or Kruskal-Wallis test when appropriate.

To explore risk factors for mortality, we employed Cox proportional hazards regression model to estimate the hazard ratios (HR) and corresponding 95% confidence intervals (CIs). We calculated the follow-up time for each individual from the date of COVID-19 diagnosis until death, 30 days if alive, or the end of the follow-up date of study (30 January 2021). Since many cases were hospitalized before COVID-19 diagnosis in the first wave, we used logistic regression to estimate the risks of the composite severity outcome, which consists of COVID-19 death and hospitalization, in odds ratio (ORs) and 95%CI. In multiple regression analyses employing either Cox or logistic regression models, we controlled demographic, socio-economic variables, LTC contacts, and all comorbidities. We conducted separate analyses for three comorbidity-related variables (whether having any comorbidity, the number of comorbidities, and types of comorbidity) by fitting them in three models to avoid collinearity. To understand the influence of comorbidities under different age groups, we conducted stratified multivariable Cox regression and logistic regression analyses to examine the associations between comorbidities and the two outcomes in five age strata: individuals below 50, between

50–59, between 60–69, between 70–79, and above 80 years old. We also applied the Kaplan-Meier product limit method to graphically examine the relation between comorbidities and time to mortality. Log-rank test was used to compare the differences in survival functions between levels of comorbidities.

All statistical analyses were performed using R version 4.0.3. All the variables had no missing variables except two, residence type and income quintile, having 0.28% missing values, which were randomly imputed based on the distributions of non-missing values of the variables. For all the statistical tests, a p-value less than 0.05 was considered statistically significant. The study is reported according to the Strengthening the Reporting of Observational Studies in Epidemiology (STROBE) guideline (S2 Appendix). Analyses were planned prior to accessing the complete data prepared by ICES, but no prospective analysis plan was published.

## Results

We identified 3,483,595 individuals who had a SARS-COV-2 tests in 2020 where we excluded 3,226,740 who had all tests as negative, and then excluded 89,335 who had a positive test after December 31, 2020. Therefore, the study included a total of 167,500 (1.1%) confirmed COVID-19 cases that were reported in Ontario, Canada as of December 31, 2020 (Fig 1). Among whom, unfortunately 2.8% (n = 4,747) deceased within 30 days after their first positive test. Of individuals with COVID-19, the average age was 42.7 years old (standard deviation, SD = 21.9), 52% (n = 87,071) were females, 24.6% (n = 41,185) belong to the lowest income quintile, and 5.6% (n = 9,357) lived in LTC facilities. The majority of individuals with COVID-19 (96.2%, n = 161,149) lived in urban areas. Nearly half (43.8%, n = 73,378) of Ontarians diagnosed with COVID-19 had at least one comorbidity. The most common comorbidities were

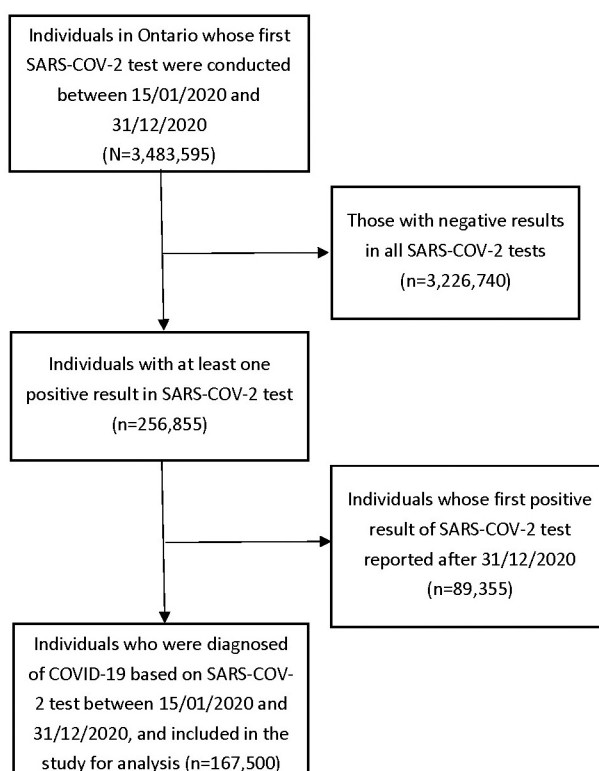

**Fig 1. Flowchart of 167,500 individuals included in this study.**

hypertension (24%, n = 40,154), asthma (16%, n = 26,814), and diabetes (14.7%, n = 24,662), while the least were HIV (0.2%, n = 332) and solid organ transplant (0.1%, n = 117, Table 1). In unadjusted analyses almost all comorbidities except HIV and inflammatory bowel disease had significant associations with mortality.

The median follow-up period under the multivariable analysis for mortality was 30 days after their first diagnoses of COVID-19. As shown in Fig 2, male sex was a significant risk factor for mortality (HR = 1.65; 95% CI 1.55–1.75; p<0.001), while older age appeared to be a strong predictor for all-cause COVID-19 mortality. Compared with those between 50–59 years old, individuals in the age groups above 80 years, between 70–79 years, and between 60–69 years had 23.10 (95%CI 19.37–27.56; p<0.001), 10.80 (95%CI 9.06–12.89; p<0.001), and 3.71 (95%CI 3.11–4.42; p<0.001) times higher risk of mortality respectively, while individuals under 50 years old had 90% lower risk of death (HR = 0.10, 95%CI 0.07–0.14, p<0.001). Individuals in higher income quintiles (quintile 4 and 5) had 13% (95%CI 0.79–0.96, p<0.001) and 14% (95%CI 0.78–0.95, p<0.001) lower risk of mortality than those in the lowest income quintile (quintile 1). Individuals who were residents in LTC facilities had 1.86 (95CI% 1.72–2.01; p<0.001) times higher risk of mortality than their counterparts who were not. Individuals with comorbidities had 2.80 times higher risk of death (95%CI 2.35–3.34; p<0.001) compared with those without, and the risk increased with respect to time (Fig 3). Furthermore, the mortality risks substantially elevated from 2.14 (95%CI 1.76–2.60) to 4.81 (95%CI 3.95–5.85) times as the number of comorbidities increased from one to five or more. Additionally, we observed higher risks of mortality in the following comorbidities: solid organ transplant (HR = 3.06, 95%CI 2.03–4.63; p<0.001), dementia (HR = 1.46, 95%CI 1.35–1.58; p<0.001), chronic kidney disease (HR = 1.45, 95%CI 1.34–1.57; p<0.001), severe mental illness (HR = 1.42, 95%CI%, 1.12–1.80; p<0.001), CVD (HR = 1.22, 95%CI, 1.15–1.30), diabetes (HR = 1.19, 95%, 1.12–1.26; p<0.001), COPD (HR = 1.19, 95%CI 1.12–1.26; p<0.001), cancer (HR = 1.17, 95%CI, 1.09–1.27; p<0.001), and hypertension (HR = 1.16, 95%CI, 1.07–1.26; p<0.001). We did not find significant associations between mortality and comorbidities such as asthma, HIV, rheumatoid arthritis, liver disease, and inflammatory bowel disease in the multivariable analysis.

In multivariable logistic models for the composite disease severity indicator (Fig 4), older age remained to be the most substantial predictor. We also observed that higher income quintile was associated with reduced odds of severe outcomes, as individuals in the highest income quintile (quintile 5) had the lowest odds (OR = 0.73, 95%CI 0.69–0.78; p<0.001) compared with individuals in lowest income quintile (quintile 1). Male sex, living in urban areas, and residents of LTC facilities remained significant predictors for disease severity. Compared with individuals without comorbidity, individuals who had comorbidities were two times more likely to experience severe outcomes (OR = 2.16, 95%CI 2.04, 2.29; p<0.001). As the number of comorbidities increased from one to five or more, the odds ratio increased substantially from 1.70 (95%CI, 1.60–1.86, p<0.001) to 6.17 (95%CI, 5.60–6.81, p<0.001). All comorbidities that were significantly associated with mortality were also found to be associated with the composite severity outcome. In addition, asthma (OR = 1.09, 95%CI 1.03–1.16, p<0.001), HIV (OR = 1.67, 95%CI 1.15–2.42, p<0.01), liver disease (OR = 1.43, 95%CI, 1.25–1.64), and rheumatoid arthritis (OR = 1.25, 95% CI 1.09–1.43; p<0.001) became significantly associated with increased odds of disease severity.

## Age stratified analysis

We conducted subgroup analyses using the multivariable Cox proportional hazard regression models to explore the effect of comorbidities on mortality in each age group. As shown in Table 2, the number of comorbidities profoundly elevated mortality risks among individuals in lower age groups. On the other hand, the influence of comorbidities on mortality reduced

**Table 1. Characteristics of individuals with COVID-19 in Ontario, Canada by December 31, 2021.**

| Characteristics | Total individuals | Alive | Deceased within 30 days after first positive COVID-19 test | P value |
|---|---|---|---|---|
| **Total (%)** | 167,500 (100) | 162,753 (97.2) | 4,747 (2.8) | |
| **Mean Age (± SD) (yrs)** | 42.7 (21.9) | 41.6 (21.1) | 82.2 (11.5) | <0.001 |
| **Age group, N (%) (yrs)** | | | | |
| < = 49 | 104716 (62.5) | 104660 (64.3) | 56 (1.2) | <0.001 |
| 50–59 | 25960 (15.5) | 25803 (15.9) | 157 (3.3) | |
| 60–69 | 16044 (9.6) | 15610 (9.6) | 434 (9.1) | |
| 70–79 | 8802 (5.3) | 7871 (4.8) | 931 (19.6) | |
| 80+ | 11978 (7.2) | 8809 (5.4) | 3169 (66.8) | |
| **Sex, N (%)** | | | | |
| Male | 80429 (48.0) | 78099 (48.0) | 2330 (49.1) | 0.14 |
| Female | 87071 (52.0) | 84654 (52.0) | 2417 (50.9) | |
| **Income quantile, N (%)** | | | | |
| 1 (lowest) | 41185 (24.6) | 39807 (24.5) | 1378 (29.0) | <0.001 |
| 2 | 36880 (22.0) | 35670 (21.9) | 1210 (25.5) | |
| 3 | 36002 (21.5) | 35080 (21.6) | 922 (19.4) | |
| 4 | 29138 (17.4) | 28501 (17.5) | 637 (13.4) | |
| 5 (highest) | 23836 (14.2) | 23250 (14.3) | 586 (12.3) | |
| Missing | 459 (0.3) | 445 (0.3) | 14 (0.3) | |
| **Rural, N (%)** | | | | |
| No | 161149 (96.2) | 156563 (96.2) | 4586 (96.6) | 0.16 |
| Yes | 5936 (3.5) | 5786 (3.6) | 150 (3.2) | |
| Missing | 415 (0.2) | 404 (0.2) | 11 (0.2) | |
| **LTC resident, N (%)** | | | | |
| No | 158143 (94.4) | 155966 (95.8) | 2177 (45.9) | <0.001 |
| Yes | 9357 (5.6) | 6787 (4.2) | 2570 (54.1) | |
| **Comorbidities** | | | | |
| **Any comorbidity, N (%)** | | | | |
| No | 94122 (56.2) | 93969 (57.7) | 153 (3.2) | <0.001 |
| Yes | 73378 (43.8) | 68784 (42.3) | 4594 (96.8) | |
| **Comorbidities, N (%)** | | | | |
| 0 | 94122 (56.2) | 93969 (57.7) | 153 (3.2) | <0.001 |
| 1 | 40141 (24.0) | 39586 (24.3) | 555 (11.7) | |
| 2 | 16363 (9.8) | 15334 (9.4) | 1029 (21.7) | |
| 3 | 8615 (5.1) | 7487 (4.6) | 1128 (23.8) | |
| 4 | 4541 (2.7) | 3680 (2.3) | 861 (18.1) | |
| 5+ | 3718 (2.2) | 2697 (1.7) | 1021 (21.5) | |
| **Asthma, N (%)** | | | | |
| No | 140686 (84.0) | 136768 (84.0) | 3918 (82.5) | 0.006 |
| Yes | 26814 (16.0) | 25985 (16.0) | 829 (17.5) | |
| **COPD, N (%)** | | | | |
| No | 157784 (94.2) | 154440 (94.9) | 3344 (70.4) | <0.001 |
| Yes | 9716 (5.8) | 8313 (5.1) | 1403 (29.6) | |
| **Dementia, N (%)** | | | | |
| No | 158890 (94.9) | 156516 (96.2) | 2374 (50.0) | <0.001 |
| Yes | 8610 (5.1) | 6237 (3.8) | 2373 (50.0) | |
| **HIV, N (%)** | | | | |

*(Continued)*

**Table 1.** (Continued)

| Characteristics | Total individuals | Alive | Deceased within 30 days after first positive COVID-19 test | P value |
|---|---|---|---|---|
| **No** | 167168 (99.8) | 162431 (99.8) | 4737 (99.8) | 0.98 |
| **Yes** | 332 (0.2) | 322 (0.2) | 10 (0.2) | |
| **Hypertension, N (%)** | | | | |
| **No** | 127346 (76.0) | 126535 (77.7) | 811 (17.1) | <0.001 |
| **Yes** | 40154 (24.0) | 36218 (22.3) | 3936 (82.9) | |
| **Diabetes, N (%)** | | | | |
| **No** | 142838 (85.3) | 140302 (86.2) | 2536 (53.4) | <0.001 |
| **Yes** | 24662 (14.7) | 22451 (13.8) | 2211 (46.6) | |
| **Chronic kidney disease, N (%)** | | | | |
| **No** | 161759 (96.6) | 158155 (97.2) | 3604 (75.9) | <0.001 |
| **Yes** | 5741 (3.4) | 4598 (2.8) | 1143 (24.1) | |
| **Cancer, N (%)** | | | | |
| **No** | 160322 (95.7) | 156580 (96.2) | 3742 (78.8) | <0.001 |
| **Yes** | 7178 (4.3) | 6173 (3.8) | 1005 (21.2) | |
| **Cardiac vascular disease, N (%)** | | | | |
| **No** | 159279 (95.1) | 156072 (95.9) | 3207 (67.6) | <0.001 |
| **Yes** | 8221 (4.9) | 6681 (4.1) | 1540 (32.4) | |
| **Rheumatoid arthritis, N (%)** | | | | |
| **No** | 165964 (99.1) | 161355 (99.1) | 4609 (97.1) | <0.001 |
| **Yes** | 1536 (0.9) | 1398 (0.9) | 138 (2.9) | |
| **Inflammatory bowel disease, N (%)** | | | | |
| **No** | 167017 (99.7) | 162285 (99.7) | 4732 (99.7) | 0.82 |
| **Yes** | 483 (0.3) | 468 (0.3) | 15 (0.3) | |
| **Liver disease, N (%)** | | | | |
| **No** | 166278 (99.3) | 161657 (99.3) | 4621 (97.3) | <0.001 |
| **Yes** | 1222 (0.7) | 1096 (0.7) | 126 (2.7) | |
| **Severe mental illness, N (%)** | | | | |
| **No** | 166384 (99.3) | 161710 (99.4) | 4674 (98.5) | <0.001 |
| **Yes** | 1116 (0.7) | 1043 (0.6) | 73 (1.5) | |
| **Solid organ transplant, N (%)** | | | | |
| **No** | 167324 (99.9) | 162600 (99.9) | 4724 (99.5) | <0.001 |
| **Yes** | 176 (0.1) | 153 (0.1) | 23 (0.5) | |

Note: 1. In the row of total the percentages are within the row, whereas for the rest of the table the percentages are in columns.

2. P values were obtained through student-t test, Wilcoxon rank-sum test, Chi-Square test, Fisher's exact test, or Kruskal-Wallis test when appropriate.

as age increases. For example, compared with individuals of the same age group without comorbidities, the mortality risk among those having five or more comorbidities was 395.44 (95%CI, 57.93–2699.44, p<0.001), 35.87 (95%CI, 18.42–69.85, p<0.001), and 12.30 (95%CI 2.94–51.46, p<0.001) times in the age groups of below 50, between 50–59, and between 60–69 years respectively. However, the associations were not significant among individuals in their 70s and 80s. Additionally, among individuals under 50 years old, elevated mortality risks were observed, including comorbidities such as HIV (HR = 13.07, 95%CI, 3.19–53.58, p<0.001), severe mental illness (HR = 13.33, 95%CI, 5.00–35.52, p<0.001), rheumatoid arthritis (HR = 6.96, 95%CI, 1.63–29.68, p<0.01), CVD (HR = 6.82, 95%CI, 2.46–18.9, p<0.001), cancer (HR = 6.82, 95%CI, 2.61–17.82, p<0.001), liver disease (HR = 6.05, 95%CI, 1.83–20.00,

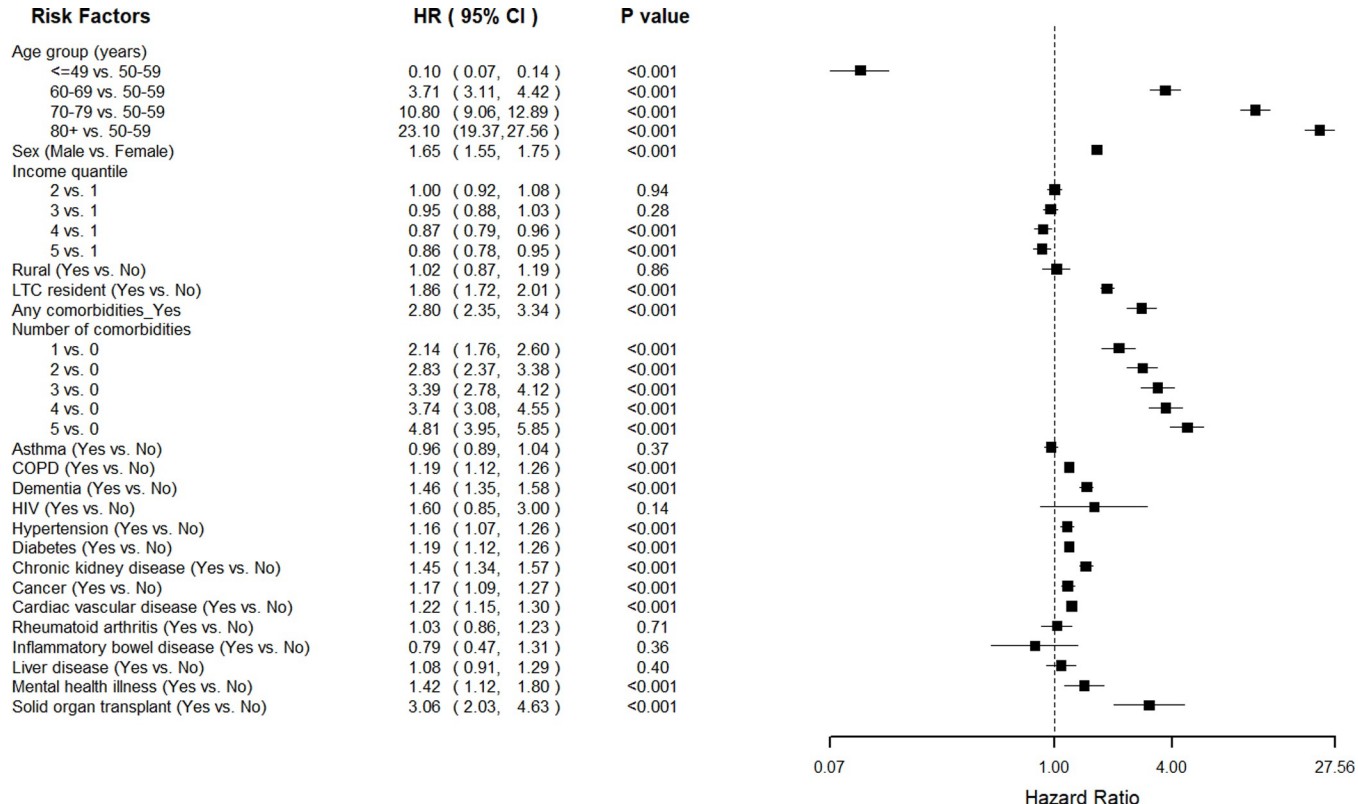

| Risk Factors | HR ( 95% CI ) | P value |
|---|---|---|
| Age group (years) | | |
| <=49 vs. 50-59 | 0.10 ( 0.07, 0.14 ) | <0.001 |
| 60-69 vs. 50-59 | 3.71 ( 3.11, 4.42 ) | <0.001 |
| 70-79 vs. 50-59 | 10.80 ( 9.06, 12.89 ) | <0.001 |
| 80+ vs. 50-59 | 23.10 (19.37, 27.56 ) | <0.001 |
| Sex (Male vs. Female) | 1.65 ( 1.55, 1.75 ) | <0.001 |
| Income quantile | | |
| 2 vs. 1 | 1.00 ( 0.92, 1.08 ) | 0.94 |
| 3 vs. 1 | 0.95 ( 0.88, 1.03 ) | 0.28 |
| 4 vs. 1 | 0.87 ( 0.79, 0.96 ) | <0.001 |
| 5 vs. 1 | 0.86 ( 0.78, 0.95 ) | <0.001 |
| Rural (Yes vs. No) | 1.02 ( 0.87, 1.19 ) | 0.86 |
| LTC resident (Yes vs. No) | 1.86 ( 1.72, 2.01 ) | <0.001 |
| Any comorbidities_Yes | 2.80 ( 2.35, 3.34 ) | <0.001 |
| Number of comorbidities | | |
| 1 vs. 0 | 2.14 ( 1.76, 2.60 ) | <0.001 |
| 2 vs. 0 | 2.83 ( 2.37, 3.38 ) | <0.001 |
| 3 vs. 0 | 3.39 ( 2.78, 4.12 ) | <0.001 |
| 4 vs. 0 | 3.74 ( 3.08, 4.55 ) | <0.001 |
| 5 vs. 0 | 4.81 ( 3.95, 5.85 ) | <0.001 |
| Asthma (Yes vs. No) | 0.96 ( 0.89, 1.04 ) | 0.37 |
| COPD (Yes vs. No) | 1.19 ( 1.12, 1.26 ) | <0.001 |
| Dementia (Yes vs. No) | 1.46 ( 1.35, 1.58 ) | <0.001 |
| HIV (Yes vs. No) | 1.60 ( 0.85, 3.00 ) | 0.14 |
| Hypertension (Yes vs. No) | 1.16 ( 1.07, 1.26 ) | <0.001 |
| Diabetes (Yes vs. No) | 1.19 ( 1.12, 1.26 ) | <0.001 |
| Chronic kidney disease (Yes vs. No) | 1.45 ( 1.34, 1.57 ) | <0.001 |
| Cancer (Yes vs. No) | 1.17 ( 1.09, 1.27 ) | <0.001 |
| Cardiac vascular disease (Yes vs. No) | 1.22 ( 1.15, 1.30 ) | <0.001 |
| Rheumatoid arthritis (Yes vs. No) | 1.03 ( 0.86, 1.23 ) | 0.71 |
| Inflammatory bowel disease (Yes vs. No) | 0.79 ( 0.47, 1.31 ) | 0.36 |
| Liver disease (Yes vs. No) | 1.08 ( 0.91, 1.29 ) | 0.40 |
| Mental health illness (Yes vs. No) | 1.42 ( 1.12, 1.80 ) | <0.001 |
| Solid organ transplant (Yes vs. No) | 3.06 ( 2.03, 4.63 ) | <0.001 |

**Fig 2. Associations of demo-socio-economic characters and comorbidities with 30-day all-cause mortality from Cox proportional hazard regression models.** Lines represent 95% confidence interval. The hazard ratios of whether having any comorbidities, and number of comorbidities were calculated based on separate multivariable Cox models that included age, sex, income quantile, rural and LTC resident.

p<0.001), and diabetes (HR = 2.48, 95%CI, 1.25–4.93, p<0.01). Among all age groups, diabetes and cancer remained a significant risk factor for mortality.

Similar patterns were identified in the subgroup analysis for the composite severity outcome that the number of comorbidities were associated with elevated risk to disease severity among younger age groups, while it became non-significant to disease severity among individuals of 80 years or older (Table 3). Among individuals under 50 years old, preexisting conditions such as severe mental illness (OR = 8.00, 95%CI 6.33–10.13, p<0.001), solid organ transplant (OR = 6.69, 95%CI 3.43–13.02, p<0.001), chronic kidney disease (OR = 2.83, 95%I 2.15–3.72, p<0.001), liver disease (OR = 2.59, 95%CI, 1.65–4.06, p<0.001), CVD (OR = 2.51, 95%CI, 1.76–3.57, p<0.001), diabetes (OR = 2.08, 95%CI, 1.81–2.38, p<0.001), cancer (OR = 1.82, 95%CI, 1.36–2.44, p<0.001), hypertension (OR = 1.45, 95%CI 1.26–1.66, p<0.001), and asthma (HR = 1.31, 95%CI, 1.16–1.47, p<0.001) were significant predicators for severe outcomes. COPD, dementia and cancer appeared significantly associated with severe outcomes in most age groups.

## Discussion

In this study, we mobilized a large population-based cohort to identify disease-related risk to 30-day all-cause mortality and severe outcomes. Between January 15 and December 31, 2020, a total of 167,500 COVID-19 cases were diagnosed in Ontario, Canada. Among the comorbidities examined in our study, we found that solid organ transplant, severe mental illness, dementia, chronic kidney disease, CVD, diabetes, hypertension, cancer and COPD were predictors of

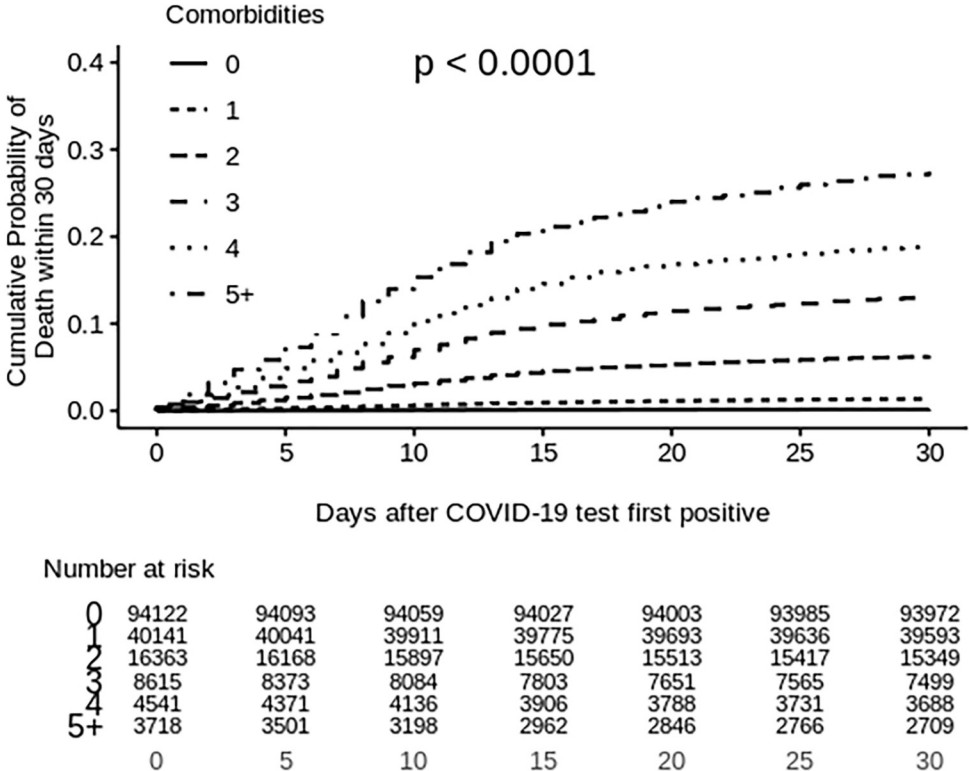

**Fig 3. Cumulative probability of all-cause death within 30 days separated by number of comorbidities.**

COVID-19 mortality and severity. We also found that increased number of comorbidities was strongly associated with COVID-19 deaths and severity, while the associations of comorbidity reduced as age increases. Comorbidities, such as HIV, severe mental illness, rheumatoid arthritis, CVD, cancer, and diabetes were associated with higher risk of COVID-19 mortality and severity among individuals under 50 years old, compared to their effect in older age groups. Our study also suggested that old age is the most significant risk factor for mortality and severity after controlling demographic and comorbidity conditions.

Our results were largely consistent with two previous population-based studies that reported CVD, cancer, liver disease, renal disease, dementia, and diabetes as risk factors to mortality [10, 14], while we added solid organ transplant, severe mental illness, hypertension and COPD to the list. Our results were also consistent with two systematic reviews synthesizing facility-based COVID-19 cohorts where CVD, hypertension, diabetes, chronic kidney disease, and cancer were identified as risk factors for COVID-19 mortality [5, 9]. Similar results were reported from a study in the United Kingdom based on hospitalized COVID-19 patients in the first wave of the pandemic [26]. We identified that solid organ transplant and severe mental illness were associated with the highest risks of mortality among all the comorbidities examined in our study. The level of risk identified in our study were similar to studies based on other large cohorts of COVID-19 patients [10, 26, 27], but was lower than the level reported from studies based on small number of patients in the beginning of the pandemic [5, 6, 9]. Similar to Harrison's study [10], we did not find that HIV was associated with mortality, probably due to the fact that we had a small number of HIV positive patients (n = 332, 0.2%). In a South African study which contained 19% HIV positive patients in its cohort, both HIV and tuberculosis appeared to be significant risk factors for mortality [12].

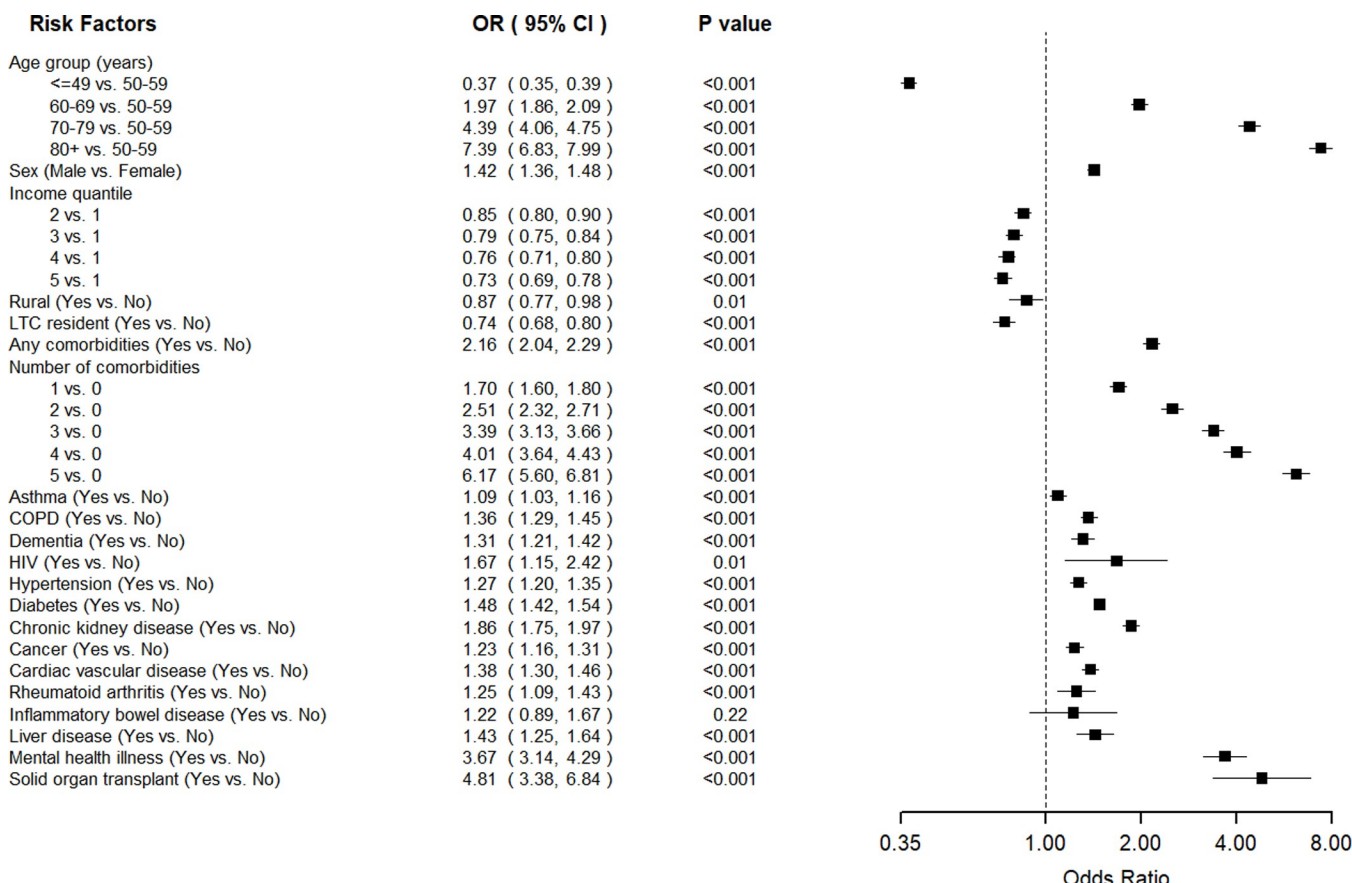

**Fig 4. Associations of demo-socio-economic characters and comorbidities with the compositive disease severity outcome that includes all-cause 30-day mortality and hospitalization from logistic regression models.** Lines represent 95% confidence interval. The hazard ratios of whether having any comorbidities, and number of comorbidities were calculated based on separate multivariable Cox models that included age, sex, income quantile, rural and LTC resident.

Our study added to the evidence base by highlighting the number of comorbidities as a strong risk factor for COVID-19 mortality and severity among younger individuals. While a previous study only showed an elevated risk of mortality among individuals with two or more comorbidities [6], we found that as the number of comorbidities increased from one to five or more, the mortality risks increased exponentially among individuals in younger age groups. On the contrary, the number of comorbidities was not a significant risk factor for mortality among individuals above 80 years old. Furthermore, we found that comorbidities had a greater impact on individuals less than 50 years old compared with those in older age groups. For individuals under 50 years old, we identified that HIV, severe mental illness, rheumatoid arthritis, CVD, cancer and diabetes posed substantially high risks of mortality or severity, which is consistent with Harrison's findings [10]. As vaccines are rolling out to younger populations, our study suggests that decision-makers should take the number of comorbidities into consideration when prioritizing high-risk groups for vaccination among young populations, instead of making decisions based solely on age groups or risk areas.

## Strengths and limitations

To our knowledge, it is one of the first studies employed near complete population-based data in examining risk factors for COVID-19 mortality and severity. Because the Government of

**Table 2. Age groups based analyses regarding associations of demo-socio-economic characters and comorbidities with all-cause 30-day mortality by Cox proportional hazards regressions.**

| Risk factor | Hazard ratio (95% CI) | | | | |
|---|---|---|---|---|---|
| | Age < = 49 years | Age 50–59 years | Age 60–69 years | Age 70–79 years | Age 80+ years |
| **Total (%)** | 104,716 (62.5) | 25,960 (15.5) | 16,044 (9.6) | 8,802 (5.3) | 11,978 (7.2) |
| **Age** (every 1-year increases) | 1.09 (1.05, 1.14) *** | 1.13 (1.06, 1.20) *** | 1.07 (1.03, 1.12) *** | 1.07 (1.05, 1.09) *** | 1.05 (1.05, 1.05) *** |
| **Sex** | | | | | |
| Male | 2.80 (1.56, 5.04) *** | 1.72 (1.23, 2.39) *** | 1.39 (1.14, 1.69) *** | 1.73 (1.51, 1.99) *** | 1.77 (1.63, 1.91) *** |
| Female | reference | reference | reference | reference | reference |
| **Income quantile** | | | | | |
| 1 (lowest) | reference | reference | reference | reference | reference |
| 2 | 0.91 (0.45, 1.85) | 0.77 (0.50, 1.19) | 0.84 (0.65, 1.09) | 0.89 (0.74, 1.06) | 1.06 (0.96, 1.17) |
| 3 | 0.86 (0.42, 1.78) | 0.76 (0.48, 1.22) | 0.76 (0.58, 1.00) | 0.86 (0.71, 1.05) | 1.06 (0.96, 1.17) |
| 4 | 0.67 (0.29, 1.56) | 0.79 (0.48, 1.28) | 0.64 (0.47, 0.88) ** | 0.80 (0.65, 1.00) | 0.96 (0.85, 1.08) |
| 5 (highest) | 0.30 (0.09, 1.04) | 0.57 (0.31, 1.05) | 0.57 (0.39, 0.83) *** | 0.80 (0.63, 1.02) | 0.97 (0.86, 1.09) |
| **Rural** | | | | | |
| No | reference | reference | reference | reference | reference |
| Yes | 1.17 (0.28, 4.91) | 0.78 (0.29, 2.12) | 1.21 (0.73, 2.01) | 0.82 (0.56, 1.19) | 1.06 (0.87, 1.29) |
| **LTC resident** | | | | | |
| No | reference | reference | reference | reference | reference |
| Yes | 10.80 (1.82, 64.30) ** | 4.71 (2.47, 9.00) *** | 3.60 (2.63, 4.92) *** | 1.97 (1.65, 2.35) *** | 1.49 (1.38, 1.61) *** |
| **Any comorbidities** | | | | | |
| No | reference | reference | reference | reference | reference |
| Yes | 3.00 (1.67, 5.41) *** | 2.39 (1.43, 3.97) *** | 2.23 (1.50, 3.29) *** | 1.86 (1.21, 2.86) *** | 1.13 (0.79, 1.60) |
| **Number of comorbidities** | | | | | |
| 0 | reference | reference | reference | reference | reference |
| 1 | 2.46 (1.37, 4.43) *** | 2.18 (1.26, 3.78) * | 1.28 (0.80, 2.06) | 1.11 (0.68, 1.80) | 1.06 (0.73, 1.54) |
| 2 | 2.39 (0.86, 6.61) | 1.65 (0.85, 3.21) | 2.05 (1.31, 3.22) *** | 1.80 (1.15, 2.83) * | 1.06 (0.75, 1.51) |
| 3 | 5.53 (1.35, 22.67) * | 4.31 (2.17, 8.55) *** | 2.66 (1.66, 4.26) *** | 2.14 (1.36, 3.36) *** | 1.11 (0.78, 1.57) |
| 4 | 20.7 (5.05, 84.88) *** | 5.05 (2.13, 11.97) *** | 5.81 (3.63, 9.30) *** | 2.25 (1.40, 3.60) *** | 1.07 (0.75, 1.53) |
| 5+ | 109.95 (33.92, 356.38) *** | 5.81 (2.27, 14.89) *** | 4.95 (2.98, 8.24) *** | 2.69 (1.68, 4.31) *** | 1.36 (0.96, 1.94) |
| **Asthma** | | | | | |
| No | reference | reference | reference | reference | reference |
| Yes | 1.12 (0.53, 2.35) | 1.42 (0.94, 2.14) | 1.08 (0.84, 1.40) | 1.02 (0.86, 1.22) | 0.96 (0.87, 1.06) |
| **COPD** | | | | | |
| No | reference | reference | reference | reference | reference |
| Yes | 1.28 (0.30, 5.48) | 1.75 (1.14, 2.69) ** | 1.26 (0.99, 1.59) | 1.30 (1.13, 1.49) *** | 1.09 (1.01, 1.18) * |
| **Dementia** | | | | | |
| No | reference | reference | reference | reference | reference |
| Yes | 0.77 (0.05, 11.09) | 2.56 (1.26, 5.18) ** | 1.82 (1.33, 2.49) *** | 1.68 (1.41, 2.01) *** | 1.28 (1.19, 1.39) *** |
| **HIV** | | | | | |
| No | reference | reference | reference | reference | reference |
| Yes | 13.07 (3.19, 53.58) *** | 0.64 (0.09, 4.66) | 1.82 (0.58, 5.68) | 0.61 (0.15, 2.44) | 1.38 (0.34, 5.54) |
| **Hypertension** | | | | | |
| No | reference | reference | reference | reference | reference |
| Yes | 0.75 (0.33, 1.70) | 0.84 (0.59, 1.20) | 1.13 (0.89, 1.43) | 1.13 (0.95, 1.35) | 0.97 (0.86, 1.09) |
| **Diabetes** | | | | | |
| No | reference | reference | reference | reference | reference |
| Yes | 2.48 (1.25, 4.93) ** | 1.92 (1.35, 2.73) *** | 1.49 (1.20, 1.85) *** | 1.30 (1.13, 1.49) *** | 1.13 (1.04, 1.22) *** |

(*Continued*)

**Table 2.** (Continued)

| Risk factor | Hazard ratio (95% CI) | | | | |
|---|---|---|---|---|---|
| | Age < = 49 years | Age 50–59 years | Age 60–69 years | Age 70–79 years | Age 80+ years |
| **Chronic kidney disease** | | | | | |
| No | reference | reference | reference | reference | reference |
| Yes | 1.60 (0.45, 5.72) | 3.56 (2.22, 5.70) *** | 2.46 (1.91, 3.17) *** | 1.42 (1.21, 1.66) *** | 1.31 (1.21, 1.42) *** |
| **Cancer** | | | | | |
| No | reference | reference | reference | reference | reference |
| Yes | 6.82 (2.61, 17.82) *** | 2.94 (1.84, 4.71) *** | 1.45 (1.10, 1.90) ** | 1.20 (1.02, 1.40) * | 1.08 (1.00, 1.17) |
| **Cardiovascular disease** | | | | | |
| No | reference | reference | reference | reference | reference |
| Yes | 6.82 (2.46, 18.90) *** | 1.36 (0.82, 2.27) | 1.82 (1.44, 2.31) *** | 1.22 (1.04, 1.43) ** | 1.11 (1.02, 1.20) ** |
| **Rheumatoid arthritis** | | | | | |
| No | reference | reference | reference | reference | reference |
| Yes | 6.96 (1.63, 29.68) ** | 1.84 (0.58, 5.85) | 1.03 (0.51, 2.09) | 1.15 (0.81, 1.64) | 1.03 (0.83, 1.28) |
| **Inflammatory bowel disease** | | | | | |
| No | reference | reference | reference | reference | reference |
| Yes | 3.29 (0.43, 25.24) | NA | 2.01 (0.65, 6.28) | 1.00 (0.32, 3.12) | 0.64 (0.32, 1.27) |
| **Liver disease** | | | | | |
| No | reference | reference | reference | reference | reference |
| Yes | 6.05 (1.83, 20.00) *** | 1.34 (0.62, 2.87) | 1.11 (0.69, 1.77) | 1.00 (0.72, 1.40) | 0.98 (0.76, 1.26) |
| **Severe mental illness** | | | | | |
| No | reference | reference | reference | reference | reference |
| Yes | 13.33 (5.00, 35.52) *** | 1.34 (0.47, 3.78) | 1.88 (1.11, 3.19) * | 0.84 (0.52, 1.38) | 1.40 (0.99, 2.00) |
| **Solid organ transplant** | | | | | |
| No | reference | reference | reference | reference | reference |
| Yes | 3.53 (0.38, 32.93) | 0.93 (0.22, 3.90) | 2.27 (1.14, 4.51) * | 2.69 (1.47, 4.94) *** | NA |

Note: 1. Statistical significance is denoted by asterisk.

* means <0.05,

** means <0.01, and

*** means <0.001.

2. Three separate models that include each of the comorbidity-related variables (i.e., whether having any comorbidity, the number of comorbidities, and types of comorbidity) were created to avoid collinearity, while this table presents the combined results.

Ontario is the single payer for all physician consultations and hospitalization under the universal health coverage, we were able to link all individuals diagnosed with COVID-19 with their previous medical history and encounters. As our database recorded exact date of mortality, we were able to report hazard ratio through Cox models, which is a more accurate estimate of risks than odds ratios [28]. As shown in S1 Appendix, the ICES chronic disease databases employed multi-database to catch the chronic disease records of all Ontarians. The majority of the chronic disease were have been validated from previous studies demonstrating high sensitivity and specificity [24, 25], so misclassification of comorbidities is minimal.

Our findings should be considered in light of a number of limitations. First, the study was designed to identify risk factors associated with mortality and severity, and the estimates reported here do not reflect any causal effects. Second, our data was collected until December 31 2020 when the new SARS-COV-2 variants of concern (VOC) including B.1.1.7 were only sporadically reported in Ontario, but the B.1.1.7 variant has become predominant in the United Kingdom and Canada at the time of writing [29]. Variant B.1.1.7 has higher

**Table 3. Age groups based analyses regarding associations of demo-socio-economic characters and comorbidities with the composite severity outcome that includes all-cause 30-day mortality and hospitalization by logistic regression models.**

| Risk factor | Odds ratio (95% CI) | | | | |
|---|---|---|---|---|---|
| | Age < = 49 years | Age 50–59 years | Age 60–69 years | Age 70–79 years | Age 80+ years |
| **Total (%)** | 104,716 (62.5) | 25,960 (15.5) | 16,044 (9.6) | 8,802 (5.3) | 11,978 (7.2) |
| **Age** (every 1-year increases) | 1.04 (1.04, 1.04) *** | 1.04 (1.02, 1.06) *** | 1.06 (1.04, 1.08) *** | 1.07 (1.05, 1.09) *** | 1.03 (1.03, 1.03) *** |
| **Sex** | | | | | |
| Male | 1.01 (0.93, 1.09) | 1.57 (1.42, 1.73) *** | 1.38 (1.25, 1.52) *** | 1.52 (1.38, 1.68) *** | 1.86 (1.72, 2.01) *** |
| Female | reference | Reference | Reference | Reference | Reference |
| **Income quintile** | | | | | |
| 1 (lowest) | reference | Reference | Reference | Reference | Reference |
| 2 | 0.81 (0.72, 0.91) *** | 0.86 (0.75, 0.99) * | 0.78 (0.69, 0.88) *** | 0.85 (0.74, 0.98) * | 0.92 (0.84, 1.02) |
| 3 | 0.75 (0.67, 0.84) *** | 0.69 (0.59, 0.81) *** | 0.70 (0.61, 0.81) *** | 0.83 (0.72, 0.95) ** | 0.96 (0.85, 1.08) |
| 4 | 0.72 (0.63, 0.82) *** | 0.79 (0.67, 0.92) *** | 0.67 (0.58, 0.77) *** | 0.73 (0.62, 0.85) *** | 0.85 (0.76, 0.96) ** |
| 5 (highest) | 0.69 (0.59, 0.81) *** | 0.73 (0.63, 0.86) *** | 0.69 (0.59, 0.81) *** | 0.75 (0.64, 0.88) *** | 0.79 (0.71, 0.89) *** |
| **Rural** | | | | | |
| No | reference | Reference | Reference | reference | Reference |
| Yes | 1.06 (0.84, 1.34) | 0.79 (0.59, 1.06) | 0.96 (0.76, 1.22) | 0.65 (0.51, 0.82) *** | 0.84 (0.68, 1.05) |
| **LTC resident** | | | | | |
| No | reference | Reference | Reference | Reference | Reference |
| Yes | 4.81 (2.52, 9.18) *** | 1.51 (1.02, 2.23) * | 1.39 (1.12, 1.73) *** | 0.66 (0.57, 0.78) *** | 0.61 (0.57, 0.66) *** |
| **Any comorbidities** | | | | | |
| No | reference | reference | reference | reference | reference |
| Yes | 1.80 (1.64, 1.99) *** | 1.63 (1.45, 1.84) *** | 1.60 (1.39, 1.84) *** | 1.72 (1.41, 2.09) *** | 0.99 (0.78, 1.25) |
| **Number of comorbidities** | | | | | |
| 0 | reference | reference | reference | reference | reference |
| 1 | 1.51 (1.37, 1.66) *** | 1.36 (1.19, 1.56) *** | 1.16 (0.99, 1.36) | 1.27 (1.02, 1.58) * | 0.92 (0.72, 1.19) |
| 2 | 2.64 (2.21, 3.15) *** | 1.62 (1.38, 1.89) *** | 1.62 (1.38, 1.89) *** | 1.62 (1.30, 2.00) *** | 0.95 (0.74, 1.23) |
| 3 | 4.85 (3.69, 6.39) *** | 2.69 (2.21, 3.27) *** | 2.05 (1.72, 2.45) *** | 1.88 (1.51, 2.33) *** | 1.00 (0.78, 1.29) |
| 4 | 6.30 (3.93, 10.08) *** | 2.64 (1.93, 3.61) *** | 2.92 (2.40, 3.55) *** | 1.92 (1.51, 2.42) *** | 0.97 (0.75, 1.25) |
| 5+ | 12.43 (6.14, 25.17) *** | 5.00 (3.59, 6.98) *** | 3.10 (2.45, 3.92) *** | 2.69 (2.13, 3.40) *** | 1.14 (0.88, 1.47) |
| **Asthma** | | | | | |
| No | Reference | reference | Reference | reference | Reference |
| Yes | 1.31 (1.16, 1.47) *** | 1.28 (1.12, 1.47) *** | 1.17 (1.04, 1.32) ** | 1.01 (0.90, 1.14) | 1.07 (0.97, 1.18) |
| **COPD** | | | | | |
| No | Reference | Reference | reference | reference | Reference |
| Yes | 1.17 (0.84, 1.64) | 1.46 (1.23, 1.74) *** | 1.52 (1.35, 1.71) *** | 1.46 (1.30, 1.64) *** | 1.12 (1.03, 1.21) ** |
| **Dementia** | | | | | |
| No | Reference | Reference | Reference | Reference | reference |
| Yes | 2.18 (0.82, 5.81) | 2.48 (1.58, 3.90) *** | 1.39 (1.08, 1.79) ** | 1.63 (1.40, 1.91) *** | 1.08 (1.00, 1.17) |
| **HIV** | | | | | |
| No | Reference | Reference | reference | Reference | reference |
| Yes | 3.06 (1.77, 5.31) *** | 0.93 (0.45, 1.93) | 0.79 (0.34, 1.88) | 0.87 (0.30, 2.51) | 2.05 (0.37, 11.53) |
| **Hypertension** | | | | | |
| No | Reference | Reference | reference | Reference | reference |
| Yes | 1.45 (1.26, 1.66) *** | 1.13 (1.02, 1.24) * | 1.09 (0.99, 1.21) | 1.11 (0.98, 1.24) | 0.95 (0.85, 1.07) |
| **Diabetes** | | | | | |
| No | reference | reference | reference | Reference | reference |
| Yes | 2.08 (1.81, 2.38) *** | 1.49 (1.33, 1.68) *** | 1.52 (1.38, 1.68) *** | 1.35 (1.22, 1.49)*** | 1.21 (1.12, 1.31) *** |

(*Continued*)

**Table 3.** (Continued)

| Risk factor | Odds ratio (95% CI) | | | | |
|---|---|---|---|---|---|
| | Age < = 49 years | Age 50–59 years | Age 60–69 years | Age 70–79 years | Age 80+ years |
| **Chronic kidney disease** | | | | | |
| No | Reference | Reference | Reference | Reference | Reference |
| Yes | 2.83 (2.15, 3.72) *** | 2.83 (2.28, 3.51) *** | 2.36 (2.02, 2.76) *** | 1.88 (1.64, 2.15) *** | 1.43 (1.30, 1.58) *** |
| **Cancer** | | | | | |
| No | Reference | Reference | Reference | Reference | Reference |
| Yes | 1.82 (1.36, 2.44) *** | 1.73 (1.40, 2.15) *** | 1.25 (1.07, 1.46) ** | 1.20 (1.06, 1.35) ** | 1.04 (0.94, 1.15) |
| **Cardiovascular disease** | | | | | |
| No | Reference | Reference | reference | reference | Reference |
| Yes | 2.51 (1.76, 3.57) *** | 1.48 (1.19, 1.83) *** | 1.62 (1.41, 1.85) *** | 1.42 (1.26, 1.60) *** | 1.19 (1.10, 1.28) *** |
| **Rheumatoid arthritis** | | | | | |
| No | Reference | Reference | Reference | Reference | Reference |
| Yes | 1.55 (0.91, 2.64) | 1.19 (0.77, 1.82) | 1.34 (0.98, 1.83) | 1.38 (1.05, 1.81) * | 1.15 (0.93, 1.43) |
| **Inflammatory bowel disease** | | | | | |
| No | Reference | Reference | Reference | reference | Reference |
| Yes | 1.72 (0.93, 3.15) | 1.80 (0.89, 3.65) | 0.88 (0.41, 1.89) | 1.70 (0.82, 3.51) | 0.68 (0.36, 1.31) |
| **Liver disease** | | | | | |
| No | Reference | Reference | Reference | reference | Reference |
| Yes | 2.59 (1.65, 4.06) *** | 1.86 (1.33, 2.59) *** | 1.26 (0.94, 1.69) | 1.19 (0.90, 1.56) | 1.04 (0.78, 1.40) |
| **Severe mental illness** | | | | | |
| No | Reference | Reference | Reference | reference | Reference |
| Yes | 8.00 (6.33, 10.13) *** | 3.35 (2.27, 4.96) *** | 2.61 (1.80, 3.79) *** | 1.09 (0.74, 1.62) | 1.75 (1.14, 2.69) ** |
| **Solid organ transplant** | | | | | |
| No | Reference | Reference | Reference | reference | reference |
| Yes | 6.69 (3.43, 13.02) *** | 1.16 (0.57, 2.35) | 5.99 (2.96, 12.13) *** | 3.35 (1.47, 7.64) *** | 0.88 (0.05, 15.36) |

Note: 1. Statistical significance is denoted by asterisk.

* means <0.05,

** means <0.01, and

*** means <0.001.

2. Three separate models that include each of the comorbidity-related variables (i.e., whether having any comorbidity, the number of comorbidities, and types of comorbidity) were created to avoid collinearity, while this table presents the combined results.

transmissibility and the younger population is more likely to be infected [30]; hence, our knowledge base regarding the risk factors for COVID-19 mortality and severity needs to be updated in light of the prevalence of the new VOCs. Third, the ICES database did not record socio-demographic data regarding ethnicity, education, and individual level income, which were found to be associated with COVID-19 outcomes in several studies. Researchers have identified that people in ethnic minority groups or with low social economic status had high risks of COVID-19 infection and death [10, 31]. Variables of ethnicity and other socio-economic status such as education may also interact with each other [32, 33]. Fourth, since the reasons of death were not available in our database, we used the all-cause 30-day mortality as our COVID-19 mortality outcome, which was also used in the previous studies [27, 34]. However, we may underestimate the risk factors as we excluded a small number of COVID-19 individuals who died more than 30 days following their diagnoses but we may also overestimate as this may include death unrelated with COVID-19. Fifth, we defined disease severity as death or hospitalization. In Ontario, Canada, all COVID-19 patients would reach a physician/ nurse

practitioner if symptoms persist and would be hospitalized if the case is severe. All Ontarian residents are covered by a tax-funded universal health coverage that removes any financial barriers to access medical care. We understand this may not be the case in other settings as patients who are hospitalized have to pay or be covered by insurance, so hospitalization may not be an appropriate indicator for severity. Sixth, most of the comorbidity information were updated until March 31 2020 except three comorbidities, cardiac heart failure, inflammatory bowel disease and rheumatoid arthritis were updated until March 31, 2019. We may miss cases diagnosed after the date. The ICES chronic disease databases do not contain further details of specific comorbidities. Seventh, although the study population is close to the entire population in Ontario it excluded a very small number of people who lived in Ontario but were not eligible for OHIP. Last but not the least, though OLIS covered almost all laboratories in Ontario, the system may slightly under report COVID-19 cases, especially at the beginning of the pandemic when laboratory testing for SARS-COV-2 was still ramping up.

In conclusion, by analysing data from a large population-based cohort that includes all COVID-19 individuals identified in 2020 in Ontario Canada, we found that solid organ transplant, severe mental illness, dementia, chronic kidney disease, CVD, diabetes, hypertension, cancer and COPD were predictors of mortality and severity. Our study highlights that the number of comorbidities was a strong risk factor for deaths and severe outcomes among the younger COVID-19 individuals. We also found that the impact of comorbidities was more substantial among individuals under 50 years old. Findings of our study suggests that in addition of prioritizing by age, vaccination priority groups should include younger population with multiple comorbidities.

## Supporting information

**S1 Appendix. List of definitions of comorbidities.**
(DOCX)

**S2 Appendix. STROBE checklist for items that should be included in reports of cohort studies.**
(DOC)

## Acknowledgments

We would thank Aasha Gnanalingam and Daniella Barron from the Institute for ICES for their assisting with data access, dataset creation, and technical support. This study was supported by the Ontario Health Data Platform (OHDP), a Province of Ontario initiative to support Ontario's ongoing response to COVID-19 and its related impacts. This study was also supported by ICES, which is funded by an annual grant from the Ontario Ministry of Health (MOH) and the Ministry of Long-Term Care (MLTC). Parts of this material are based on data and information compiled and provided by the Canadian Institute for Health Information (CIHI) and the MOH. The analyses, conclusions, opinions and statements expressed herein are solely those of the authors and do not reflect those of ICES, the funding or data sources; no endorsement is intended or should be inferred. No endorsement by the OHDP, its partners, or the Province of Ontario is intended or should be inferred.

## Author Contributions

**Conceptualization:** Erjia Ge, Xiaolin Wei.

**Formal analysis:** Yanhong Li.

**Funding acquisition:** Xiaolin Wei.

**Validation:** Erjia Ge, Xiaolin Wei.

**Writing – original draft:** Erjia Ge, Xiaolin Wei.

**Writing – review & editing:** Yanhong Li, Shishi Wu, Elisa Candido.

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
