## [Decision Letter · Decision Letter 0]

13 Aug 2021

PONE-D-21-19230

Association of pre-existing comorbidities with mortality and disease severity among 167,500 individuals with COVID-19 in Canada: a population-based cohort study

PLOS ONE

Dear Dr. Wei,

Thank you for submitting your manuscript to PLOS ONE. After careful consideration, we feel that it has merit but does not fully meet PLOS ONE’s publication criteria as it currently stands. Therefore, we invite you to submit a revised version of the manuscript that addresses the points raised during the review process.

We look forward to receiving your revised manuscript.

Kind regards,

Orvalho Augusto, MD, MPH

Academic Editor

PLOS ONE

b) If there are no restrictions, please upload the minimal anonymized data set necessary to replicate your study findings as either Supporting Information files or to a stable, public repository and provide us with the relevant URLs, DOIs, or accession numbers. For a list of acceptable repositories, please see http://journals.plos.org/plosone/s/data-availability#loc-recommended-repositories

Additional Editor Comments (if provided):

This is an important and well-written report to help set action priorities to mitigate COVID-19 consequences. It is one of the few population based cohort analysis. The authors use multiple health administrative databases covering the Ontario population to identify SARS-CoV-2 positive cases, build a retrospective cohort of SARS-CoV-2 infected patients, and ascertain all-cause mortality and hospitalization within 30 days post-test. The period they cover (January to December 2020) is largely prior to vaccination availability.

Few issues to be addressed:

1. Create a specific ethics consideration subsection

2. The definition of severity here is a composite of hospitalization or death within 30 days. This is problematic as one of the reviewers point out. Please discuss this in the limitations.

3. Statistical analysis subsection: Lines 177 and 178, it is written that no model including the three comorbidity-related variables (whether having any comorbidity, the number of comorbidities, and types of comorbidity) was created due to potential colinearity. However, table 2 includes such kind of model. Can you clarify this?

4. Statistical analysis subsection: line 187. There is confusion between “R” and “R Studio”. Can you please correct to cite R.

5. Line 202 put the age unit.

6. Table 1 - Add a row to put totals rather than putting them on the header. In this row you can, for example, put 167500 (100) which will alert that percentages are in columns

7. Table 2 - Add a row for total participants in the analysis in each column. And make sure the rows are well aligned (for example, the number of comorbidities is quite hard to follow).

8. Figure 2 - please label the X-axis to say “HR” or “Hazard-ratio”

9. Figure 3 - it would be better to use on X-axis multiples of 7 (0, 7, 14, 21, 28) and 30 days because most of the descriptions use weeks. This is a suggestion.

In fact, this figure suggests different hazards pre-day 15 compared to post-day 15. Did you assess for time-varying coefficients in this analysis? And did you assess the proportionality assumption?

10 . Figure 4 - This plot is based on logistic regression, right? So change the “HR” to “OR”. And label the X-axis to mean either “Odds ratio” or “OR”. Also, change the footnote (lines 268 to 270).

Reviewers' comments:

Reviewer's Responses to Questions

**Comments to the Author**

1. Is the manuscript technically sound, and do the data support the conclusions?

Reviewer #1: Yes

Reviewer #2: Partly

2. Has the statistical analysis been performed appropriately and rigorously? 

Reviewer #1: Yes

Reviewer #2: Yes

3. Have the authors made all data underlying the findings in their manuscript fully available?

Reviewer #1: Yes

Reviewer #2: Yes

4. Is the manuscript presented in an intelligible fashion and written in standard English?

Reviewer #1: Yes

Reviewer #2: Yes

5. Review Comments to the Author

Reviewer #1: The paper deals with an important topic. In their study, the authors have demonstrated that the risk of death among COVID-19 patients, increases with the number of co-morbidities, at all ages. The findings are not really new, but it’s important that they be documented.

The study design is appropriate and the methodology appears sound.

I believe there should be more discussion on potential biases, particularly information bias on the co-morbidities. These could affect the strength of the findings. Also a discussion of attributable risk of the major co-morbidities could be useful.

Reviewer #2: This is a retrospective cohort study of 167,500 individuals diagnosed of COVID-19 in Ontario throughout the 12 months of 2020 which aimed to examine the associations of comorbidities with mortality and disease severity in individuals with COVID-19. The results obtained are very relevant and consistent with two previous population-based studies and added other four comorbidities (solid organ transplant, severe mental illness, hypertension and COPD) to the list of risk factors.

For a better understanding of the manuscript these comments are formulated

Comments:

1. The study is not really designed to measure the severity of COVID-19. The title of the manuscript should not contain the words “disease severity” since the secondary outcome in the study do not include the usual items of “disease severity” (such as requirement of non-invasive ventilation, mechanical ventilation or UCI admission) and instead it only includes hospitalization which is not a criterion of severity by itself. For the same reason, the terms "disease severity" should not be used throughout the text of the manuscript.

2. Methods. The ICES database did not record socio-demographic data regarding ethnicity, education, and individual level income, which were found to be associated with COVID-19 outcomes in several studies. However, the manuscript says "we employed Cox proportional hazards regression models and logistic regression models to adjust for demographic, socio-economic variables…” This apparent contradiction should be clarified and corrected

3. The definition of comorbidities that are risks factors for mortality should be better specified. For example, how the presence of “severe mental illness” , “human immunodeficiency virus infection” , “cancer” or “rheumatoid arthritis” have been defined. In HIV patients it is very important to know whether or not they are in virological remission and immune recovery. It is also necessary to know whether cancer patients have a disseminated disease or are in partial or complete remission, or whether or not patients with rheumatoid arthritis are receiving corticosteroids or immunosuppressive medications.

4. In the Discussion, lines 338-340, the sentence “Asthma, HIV, and rheumatoid arthritis were significantly associated with severity but not deaths, indicating the three conditions were more related to COVID-19 hospitalization” should be changed .

5. In conclusion paragragh, the sentence: “the number of comorbidities was a strong risk factor for deaths and severe outcomes among the younger COVID-19 individuals” should be changed since severity outcomes were not actually measured

6. Also, in the conclusion paragraph, the sentence “Findings of our study suggests that in addition of prioritizing by age, vaccination priority groups should include younger population with multiple comorbidities” must be modified because it is an interpretation that is not derived directly from the results of the study

7.

6. PLOS authors have the option to publish the peer review history of their article (what does this mean?). If published, this will include your full peer review and any attached files.

Reviewer #1: **Yes: **Manfred S. Green

Reviewer #2: No

---

## [Author Response · Author response to Decision Letter 0]

7 Sep 2021

1. Create a specific ethics consideration subsection

Answer: Thanks. We have created a new subsection ethics consideration in Methods.

The study has received ethical approval from the Office of Research Ethics at the University of Toronto (#39138). The consent form was not obtained as the data is from ICES, which is an independent, non-profit research institute whose legal status under Ontario’s health information privacy law allows it to collect and analyze health care and demographic data, without consent, for health system evaluation and improvement.

2. The definition of severity here is a composite of hospitalization or death within 30 days. This is problematic as one of the reviewers point out. Please discuss this in the limitations.

Answer: We defined disease severity of COVID-19 based on previous studies on similar topic. As a novel pandemic, there has not been consistent definitions for COVID severity. We employed this definition because in Ontario, Canada, all cases have been reviewed by physicians and only severe cases are hospitalized. In addition, under the Canadian Medicare all Ontario residents are entitled to free consultation and hospitalization, so there are no accessibility issues. We revised in the limitation.

3. Statistical analysis subsection: Lines 177 and 178, it is written that no model including the three comorbidity-related variables (whether having any comorbidity, the number of comorbidities, and types of comorbidity) was created due to potential colinearity. However, table 2 includes such kind of model. Can you clarify this?

Answer: Thanks for pointing this out. We conducted separate models but combined the results into one table to avoid multi tables. We added one footnote under each of the tables to avoid the confusion. 

Three separate models that include each of the comorbidity-related variables (i.e., whether having any comorbidity, the number of comorbidities, and types of comorbidity) were created to avoid collinearity, while this table presents the combined results. 

4. Statistical analysis subsection: line 187. There is confusion between “R” and “R Studio”. Can you please correct to cite R.

Answer: Thanks. We changed to R.

5. Line 202 put the age unit.

Answer: Yes, it was years.

6. Table 1 - Add a row to put totals rather than putting them on the header. In this row you can, for example, put 167500 (100) which will alert that percentages are in columns

Answer: Thanks. We revised Table 1.

7. Table 2 - Add a row for total participants in the analysis in each column. And make sure the rows are well aligned (for example, the number of comorbidities is quite hard to follow).

Answer: Thanks. We added the row and also realigned the table. 

8. Figure 2 - please label the X-axis to say “HR” or “Hazard-ratio”

Answer: Thanks. We revised.

9. Figure 3 - it would be better to use on X-axis multiples of 7 (0, 7, 14, 21, 28) and 30 days because most of the descriptions use weeks. This is a suggestion.

In fact, this figure suggests different hazards pre-day 15 compared to post-day 15. Did you assess for time-varying coefficients in this analysis? And did you assess the proportionality assumption?

Answer: 

Thank you for the suggestion. We used days to describe deaths and other outcomes instead of weeks to conduct the analysis, so it does not specifically give a break by weeks. Using 7 day as a break will make it hard for us to divide the x- axis as the total is 30 days (then will have two days left). Thus, it is better to use 5 days break. 

As the data did not have comorbidity diagnosis date variables, and all the risk factors were occurred before the diagnosis of COVID-19, so we did not do time-varying analysis. 

For the Cox models, we checked proportional assumption for each age sub-cohort. There was assumption violation in a sub-cohort, but the models were robust to the violation. So, we did not stratify the variables that violated the assumption. The overall proportional assumptions were met in all the other sub-cohorts. 

10 . Figure 4 - This plot is based on logistic regression, right? So change the “HR” to “OR”. And label the X-axis to mean either “Odds ratio” or “OR”. Also, change the footnote (lines 268 to 270).

Answer: Thank you so much for letting us know the typo! We corrected it and added X labels “Odds Ratio” under Figure 4.

Reviewer #1: The paper deals with an important topic. In their study, the authors have demonstrated that the risk of death among COVID-19 patients, increases with the number of co-morbidities, at all ages. The findings are not really new, but it’s important that they be documented.

The study design is appropriate and the methodology appears sound.

I believe there should be more discussion on potential biases, particularly information bias on the co-morbidities. These could affect the strength of the findings. Also a discussion of attributable risk of the major co-morbidities could be useful.

Answer: Thanks for the suggestion. We added this in the discussion:

As shown in Appendix 1, the ICES chronic disease databases employed multi-database to catch the chronic disease records of all Ontarians. The majority of the chronic disease has been validated from previous studies demonstrating high sensitivity and specificity [24, 25], so misclassification of comorbidities is minimal.

For the attributable risk of comorbidities, the data shown that, among the total 4747(= 4594+153) death cases, including the study individuals with comorbidities and without comorbidities, more than 92% of them were attributable to comorbidities. However, please keep in mind that, this is just like an univariable analysis as the attributable risk does not being adjusted other confounders, so I think it will be misleading if presenting the result of attributable risk.

Reviewer #2: This is a retrospective cohort study of 167,500 individuals diagnosed of COVID-19 in Ontario throughout the 12 months of 2020 which aimed to examine the associations of comorbidities with mortality and disease severity in individuals with COVID-19. The results obtained are very relevant and consistent with two previous population-based studies and added other four comorbidities (solid organ transplant, severe mental illness, hypertension and COPD) to the list of risk factors.

For a better understanding of the manuscript these comments are formulated

Comments:

1. The study is not really designed to measure the severity of COVID-19. The title of the manuscript should not contain the words “disease severity” since the secondary outcome in the study do not include the usual items of “disease severity” (such as requirement of non-invasive ventilation, mechanical ventilation or UCI admission) and instead it only includes hospitalization which is not a criterion of severity by itself. For the same reason, the terms "disease severity" should not be used throughout the text of the manuscript.

Answer: As a novel pandemic, there has not been consistent definitions for COVID severity. We employed this definition because in Ontario, Canada, all cases have been reviewed by physicians and only severe cases are hospitalized. In addition, under the Canadian Medicare all residents in Ontario are entitled to free consultation and hospitalization, so there is no accessibility issue for medical care. In another word, patients who did not have severe symptoms are not hospitalized in Ontario. We also have information regarding ICU admission in the database but we feel that hospitalization for COVID itself means a lot to patients, that can be justified as a severe outcome. We understand this may not be the case in other settings as patients who are hospitalized have to pay or be covered by insurance, so hospitalization may not be an appropriate indicator for severity. We mentioned this in the limitation.

2. Methods. The ICES database did not record socio-demographic data regarding ethnicity, education, and individual level income, which were found to be associated with COVID-19 outcomes in several studies. However, the manuscript says "we employed Cox proportional hazards regression models and logistic regression models to adjust for demographic, socio-economic variables…” This apparent contradiction should be clarified and corrected

Answer: The ICES database contains the income level as defined below. Statistics Canada has developed a composite indicator that takes account of the income, household size and cost of living in the province. In this way, we felt that we did control socioeconomic indicators. But we certainly did not have ethnicity and education level. We revised the limitation.

The income variable was neighborhood-based and determined using methods developed by Statistics Canada, where income was adjusted for household size and cost of living across the province so that each dissemination area would have 20% of its population in each income quintile. Quintile five indicated the highest income group while quintile one indicated the lowest. Each individual was assigned with the neighborhood income quintile of the dissemination area which was matched with his/her postal code [20].

3. The definition of comorbidities that are risks factors for mortality should be better specified. For example, how the presence of “severe mental illness” , “human immunodeficiency virus infection” , “cancer” or “rheumatoid arthritis” have been defined. In HIV patients it is very important to know whether or not they are in virological remission and immune recovery. It is also necessary to know whether cancer patients have a disseminated disease or are in partial or complete remission, or whether or not patients with rheumatoid arthritis are receiving corticosteroids or immunosuppressive medications.

Answer: Please see the detailed definition of each comorbidity in Appendix 1. The ICES chronic disease databases do not contain further details. We also add this into the limitation.

4. In the Discussion, lines 338-340, the sentence “Asthma, HIV, and rheumatoid arthritis were significantly associated with severity but not deaths, indicating the three conditions were more related to COVID-19 hospitalization” should be changed .

Answer: Thanks. We also felt that this statement is probably misleading and deleted it.

5. In conclusion paragragh, the sentence: “the number of comorbidities was a strong risk factor for deaths and severe outcomes among the younger COVID-19 individuals” should be changed since severity outcomes were not actually measured

Answer: Please see Table 3 where associations with COVID severity by age groups were presented.

6. Also, in the conclusion paragraph, the sentence “Findings of our study suggests that in addition of prioritizing by age, vaccination priority groups should include younger population with multiple comorbidities” must be modified because it is an interpretation that is not derived directly from the results of the study

Answer: Please see the above. The study did measure associations with both severity and deaths, and presented the results in different tables (Table 3 and Table 4).

---

## [Decision Letter · Decision Letter 1]

20 Sep 2021

Association of pre-existing comorbidities with mortality and disease severity among 167,500 individuals with COVID-19 in Canada: a population-based cohort study

PONE-D-21-19230R1

Dear Dr. Wei,

We’re pleased to inform you that your manuscript has been judged scientifically suitable for publication and will be formally accepted for publication once it meets all outstanding technical requirements.

Kind regards,

Orvalho Augusto, MD, MPH

Academic Editor

PLOS ONE

Additional Editor Comments (optional):

This is the revised version of a very important report of a population-based cohort study of SARS-CoV-2 positive patients.

Few general comments: Please make sure the abbreviation SARS-CoV-2 is written the same in the document. You do have a mixture of SARS-COV-2 and others SARS-CoV-2.

Abstract: No comments

Author’s summary:

- No limitation is included here. Please add this.

- Line 91 - multivariate regression. I would suggest changing this to “multiple regression” or “multivariable regression”

Background: No comments

Methods:

- A minor issue is that the income variable has the issue the quintile definition is specific to each dissemination area. So two different dissemination areas would be seen the same.

- Line 231 - change the “multivariate analyses” to “multiple regression analyses” or “multivariable regression analyses”.

- Thank you for adding the STROBE statement form.

Results:

- Table 1 - a) please add a footnote to alert the reader that in the rows of totals the percentages are within the row, whereas for the rest of the table the percentages are in columns; b) state how the p-values were computed

- Lines 272, 296, 306, 330 change the “multivariate” to “multivariable”.

Reviewers' comments:

Reviewer's Responses to Questions

**Comments to the Author**

1. If the authors have adequately addressed your comments raised in a previous round of review and you feel that this manuscript is now acceptable for publication, you may indicate that here to bypass the “Comments to the Author” section, enter your conflict of interest statement in the “Confidential to Editor” section, and submit your "Accept" recommendation.

Reviewer #1: All comments have been addressed

Reviewer #2: All comments have been addressed

2. Is the manuscript technically sound, and do the data support the conclusions?

Reviewer #1: (No Response)

Reviewer #2: Yes

3. Has the statistical analysis been performed appropriately and rigorously? 

Reviewer #1: (No Response)

Reviewer #2: Yes

4. Have the authors made all data underlying the findings in their manuscript fully available?

Reviewer #1: (No Response)

Reviewer #2: Yes

5. Is the manuscript presented in an intelligible fashion and written in standard English?

Reviewer #1: (No Response)

Reviewer #2: Yes

6. Review Comments to the Author

Reviewer #1: All the comments in the review have been addressed. The paper reads well and the limitations of the study have been clarified.

Reviewer #2: The key points raised have been solved. Therefore the study indicates that the number of comorbidities was a strong risk factor for deaths and severe outcomes among the younger COVID-19 patients

7. PLOS authors have the option to publish the peer review history of their article (what does this mean?). If published, this will include your full peer review and any attached files.

Reviewer #1: No

Reviewer #2: No

---

## [Editor Report · Acceptance letter]

24 Sep 2021

PONE-D-21-19230R1 

Association of pre-existing comorbidities with mortality and disease severity among 167,500 individuals with COVID-19 in Canada: a population-based cohort study 

Dear Dr. Wei:

I'm pleased to inform you that your manuscript has been deemed suitable for publication in PLOS ONE. Congratulations! Your manuscript is now with our production department. 

Kind regards, 

on behalf of

Dr. Orvalho Augusto 

Academic Editor

PLOS ONE